# Macroinvertebrate Response to Internal Nutrient Loading Increases in Shallow Eutrophic Lakes

**DOI:** 10.3390/biology12091247

**Published:** 2023-09-18

**Authors:** Kai Peng, Rui Dong, Boqiang Qin, Yongjiu Cai, Jianming Deng, Zhijun Gong

**Affiliations:** 1Taihu Laboratory for Lake Ecosystem Research, State Key Laboratory of Lake Science and Environment, Nanjing Institute of Geography and Limnology, Chinese Academy of Sciences, 73 East Beijing Road, Nanjing 210008, China; kpeng@niglas.ac.cn (K.P.); dongrui21@mails.ucas.ac.cn (R.D.); qinbq@niglas.ac.cn (B.Q.); jmdeng@niglas.ac.cn (J.D.); zjgong@niglas.ac.cn (Z.G.); 2University of Chinese Academy of Sciences, Beijing 100049, China; 3School of Geography & Ocean Science, Nanjing University, 163 Xianlin Street, Nanjing 210023, China; 4Collaborative Innovation Center of Recovery and Reconstruction of Degraded Ecosystem in Wanjiang Basin Co-Founded by Anhui Province and Ministry of Education, Anhui Normal University, Wuhu 241000, China

**Keywords:** hypoxia, community structure, oligochaetes, Lake Taihu

## Abstract

**Simple Summary:**

Internal loading can significantly delay the improvement of eutrophication in lakes. When external nutrient inputs are controlled and internal loading intensifies, the dissolved oxygen condition at the water-sediment interface improves during non-algal bloom seasons, but more anaerobic conditions occur during algal bloom seasons. Macroinvertebrates, living at the water-sediment interface, are particularly sensitive to such changes. Our results indicate that, in large shallow Lake Taihu, the changes in dissolved oxygen conditions at the water-sediment interface, caused by intensified internal loading, significantly affect the community structure, diversity, and response to eutrophication of macroinvertebrates.

**Abstract:**

In eutrophic lakes, even if external loading is controlled, internal nutrient loading delays the recovery of lake eutrophication. When the input of external pollutants is reduced, the dissolved oxygen environment at the sediment interface improves in a season without algal blooms. As an important part of lake ecosystems, macroinvertebrates are sensitive to hypoxia caused by eutrophication; however, how this change affects macroinvertebrates is still unknown. In this study, we analysed the monitoring data of northern Lake Taihu from 2007 to 2019. After 2007, the external loading of Lake Taihu was relatively stable, but eutrophication began to intensify after 2013, and the nutrients in the sediments also began to decline, which was related to the efficient use of nutrients by algal blooms. The community structure and population density of macroinvertebrates showed different responses in different stages. In particular, the density of oligochaetes and the Shannon–Wiener index showed significant differences in their response to different stages, and their sensitivity to eutrophication was significantly reduced. Under eutrophication conditions dominated by internal loading, frequent hypoxia occurs at the sediment interface only when an algal bloom erupts. When there is no bloom, the probability of sediment hypoxia is significantly reduced under the disturbance of wind. Our results indicate that the current method for evaluating lake eutrophication based on oligochaetes and the Shannon–Wiener diversity index may lose its sensitivity.

## 1. Introduction

Excessive accumulation of nutrients resulting from human activities has led to eutrophication, posing significant threats to lake ecosystems. Internal loading is an important source of nutrients in lake water. Even in some lakes, more than half of the nutrients that flow into a lake through surface runoff will pass through the nutrient cycle and then enter the sediments to be buried [1]. High internal phosphorus loading from lake sediments is frequently considered an important factor delaying lake recovery after a reduction in external loading [2,3]. Many factors (such as resuspension, temperature, redox, pH, and iron:phosphorus ratio) have been proposed to be responsible for the release of phosphorus from lake sediments [4]. Especially in shallow lakes, under the influence of wind disturbance, sediment nutrients are continuously released into the lake water column [5]; these nutrients then promote phytoplankton growth. According to Wu et al. [6], the contribution of internal loading to the change in nutrients in lake water is greater than that of external inputs in Lake Dianchi, China.

Benthic macroinvertebrates mainly live on the surfaces of sediments, and changes in the physical and chemical properties of sediments impact them [7]. Macroinvertebrates are sensitive to eutrophication [8], and a variety of methods based on macroinvertebrates have been developed to assess a lake’s trophic status [9,10,11,12]. Eutrophication affects the community structure, leading to a reduction in diversity and an increase in the number of pollution-tolerant species [13,14,15], in part due to large fluctuations in oxygen at the water-sediment interface [16,17]. The more severe the eutrophication is, the more likely the water-sediment interface is to become anoxic [18]. Despite this, few studies have investigated changes in macroinvertebrates when internal loading is increased. Increased internal loading leading to eutrophication will result in hypoxia at the water-sediment interface during algal bloom seasons. However, during non-algal bloom seasons, the concentration of dissolved oxygen at the water-sediment interface will improve due to the control of external pollutants. This is a common phenomenon that has occurred in many lakes [19], such as Lake Erie [20] and Lake Taihu, and internal loading is considered to be the main reason [1,21].

Under climate change scenarios, global warming [22], wind speed decline [23], and typhoon transit [24] will aggravate internal release, which in turn leads to increased eutrophication. Therefore, in the future, there will be increasing eutrophication dominated by internal loading. By analysing the changes in macroinvertebrates to eutrophication dominated by internal loading, it is not only helpful to understand the response of lake ecosystems to eutrophication dominated by internal release, but it can also help evaluate the effectiveness of eutrophication assessment based on macroinvertebrates.

In China, due to rapid economic and population growth, the water pollution caused by the discharge of pollutants has gradually increased, especially eutrophication. Subsequently, the Chinese government made many investments in water pollution control and set up the Major Science and Technology Program for Water Pollution Control and Treatment in 2006 to develop pollution control technologies. After nearly ten years of management, nutrients in major rivers and lakes have been reported to have declined [25,26], and the algal bloom magnitudes in lakes began to decline in 2006 but rebounded after 2014 [26]. For instance, in 2007, a drinking water crisis occurred in Lake Taihu [27]. In response to this event, the government implemented several control measures aimed at reducing nutrient loading and ensuring the safety of drinking water. These measures have succeeded; however, algal blooms in Lake Taihu continue to rebound. Previous studies have shown that this may be caused by climate change [1,23]. For example, increases in temperature and decreases in wind speed can both contribute to the intensification of internal loading in Lake Taihu [23]. Undoubtedly, in this context, the proportion of nutrients provided by internal loading for the growth of cyanobacteria will increase.

For lakes, the reduction in organic-pollutant input will improve the anoxic conditions of the water-sediment interface, and during the eruption of cyanobacteria blooms, the anoxic conditions of the water-sediment interface will be aggravated. Hypoxia is fatal to lake macroinvertebrates, and how macroinvertebrates respond to this change has not yet been studied.

In this study, the northern bay area of Lake Taihu (hereafter called “north Taihu”) was taken as the research object. This lake is one of the lakes that has received the most limnological research [28] and has had a large number of field observations and experimental studies [23,29]. It has been confirmed that many factors have caused the lake’s internal release to increase [1]. We analysed 13 years of monitoring data (2007–2019) for macroinvertebrates, sediment nutrients, and water quality. According to the degree of cyanobacteria blooms in Lake Taihu that began to rebound in 2014, northern Taihu was divided into two stages: eutrophication dominated by external input and eutrophication dominated by internal loading. We analysed how macroinvertebrates responded during these two stages. We have made the following hypothesis: (1) Due to the different changes in water-sediment interface hypoxia in different seasons, the sensitivity of macroinvertebrate responses is weaker than that of eutrophication dominated by exogenous inputs, (2) If hypothesis 1 is valid, then the sensitivity of using macroinvertebrates for eutrophication assessment, based on their correlation with eutrophication, would likely be compromised.

## 2. Materials and Methods

### 2.1. Study Area and Sampling Sites

Lake Taihu, the third largest freshwater lake in China (Figure 1), is a large (2338.1 km^2^), shallow (mean depth = 1.9 m, max depth = 3.4 m), and eutrophic lake located in the southern Yangtze River Delta [30]. To better analyse the relationship between the macroinvertebrate community structure and eutrophication in Taihu, the areas with more severe eutrophication in the north were selected as the research areas. The external pollution of Taihu mainly comes from river inputs in the north, so the northern bay area is directly affected by external pollution. The sediment and water quality in North Taihu have strong spatial homogeneity, and the macroinvertebrate community structure and density between 9 different sampling sites are similar. Among them, 9 sampling sites were selected in North Taihu. The sampling times were February, May, August, and November of 2007–2019 (representing winter, spring, summer, and autumn, respectively).

Macroinvertebrate samples were collected with a modified Peterson grab (0.025 m^2^ sampling area, three replicates) and sieved in situ through 250-μm-aperture mesh. In the laboratory, the samples were sorted on a white tray, and the specimens were preserved in a 7% buffered formalin solution. The specimens were identified to the species or genus level and counted. Water temperature (Wtemp), depth, and pH were measured in situ by a multiparameter water-quality sonde (YSI6600, Yellow Springs, OH, USA), and transparency (SD) was measured by a Secchi disk.

During invertebrate sampling, mixed water samples from the surface layer, middle layer, and bottom layer were collected at each site. The total nitrogen (TN), total phosphorus (TP), chlorophyll (Chla), and permanganate index (COD_Mn_) concentrations were measured in the laboratory based on [31]. In Lake Taihu, to obtain information about the substrate, surface sediment samples were collected using modified Peterson grabs during annual invertebrate sampling in May and analysed in the laboratory for loss on ignition (LOI), sediment total nitrogen (TNs), sediment total phosphorus (TPs), and particle size. Since the sediments in North Taihu are mainly clay and silt, their proportions change in the opposite trend, so the following analysis only selects the percentage of silt (% silt) to characterize the change in particle size. Sediments were air-dried and ground into powder until they could pass through a 150-μm mesh screen, and a 0.02 g aliquot of powder was put into 25 mL of deionized water to determine sediment TN and TP with the same method as that used for the water samples. LOI was measured at 550 °C for 2 h. The particle size was measured using a laser particle size analyser (Mastersizer 2000, Malvern Instruments Ltd., Malvern, UK). Fish predation pressure adopts the fish yield data of Taihu from 2007 to 2019.

High-frequency monitoring of the dissolved oxygen concentration at the bottom of Lake Taihu (10 cm above the water-sediment interface) was conducted every hour using YSI 6600 V2 multisensor sondes (Yellow Springs Instruments, Yellow Springs, OH, USA). The monitoring period spanned from 10 October 2007 to 15 July 2016, and the sensors were deployed in Meiliang Bay (31°25′ N, 120°13′ E). Daily wind data from 1 January 2007 to 31 December 2020 from the WuXi weather station (No. 58354, 31°36′ N, 120°21′ E), located near Meiliang Bay and Zhushan Bay, were obtained from the China Meteorological Data Sharing Service System (http://cdc.cma.gov.cn (accessed on 10 November 2022)).

The geom_smooth function [32] in ggplot2 was used to determine the long-term trends of the macroinvertebrate density change and to mark the 95% confidence interval. Mann–Kendall trend analyses (MK test) were used to test their interannual trends. MK tests were carried out using the function “Kendall” in the R package Kendall [33]. Turning point analyses were used to identify the year in which a significant change in macroinvertebrate density occurred. Turning point analyses were carried out using the function “turnpoints” in the R package pastecs [34].

### 2.2. Eutrophication Evaluation

The trophic level index (TLI) recommended by the Chinese National Environment Monitoring Center was used to assess the eutrophication level of lakes. This index was assessed using five main indicators of responses to nutrients: Chla, TP, TN, SD, and COD_Mn_ [35,36]. According to the results of previous studies, the benthic oligochaete abundance and the Shannon–Wiener index have a high correlation with the lake trophic level [37,38], and many assessment methods that use macroinvertebrates to test lake eutrophication pressure also consider the response of oligochaete numbers and the diversity index [11,21]. Therefore, in this study, two indicators, oligochaete density and the Shannon–Wiener index, were selected to analyse the relationship between benthic macroinvertebrates and the level of eutrophication. To minimize the sampling error of macroinvertebrates, the average density value of each lake was determined.

Regression analysis was performed to analyse the relationship between the density and diversity index of benthic macroinvertebrates and eutrophication. Before performing regression analysis, we used an empirical exponent of 0.25 to transform the macroinvertebrate density data to better approximate a normal distribution [39]. Correlation analysis and regression analysis were performed in R. Differences were reported as significant if *p* < 0.05. For the calculation of the Shannon-Wiener diversity index, Margalef’s diversity index and Pielou’s index were used; see Shannon [40], Margalef [41], and Pielou [42].

### 2.3. Statistical Analysis

We used nonmetric multidimensional scaling (NMDS) analysis to show the characteristics of the macroinvertebrate community changes. In the multivariate analysis, species density data were used, and the top 20 dominant species were selected. Since the sediment is sampled only in May every year, the macroinvertebrate data in May were used for analysis. NMDS analysis based on Bray–Curtis distances produced three-dimensional solutions meeting the criterion of a final stress of <0.2 [32]. The NMDS analysis was employed using the R package ‘vegan’ [43]. In NMDS analysis, environmental variables were fitted by vector fitting into the configuration to show the direction of the most rapid change in the environmental variable within the ordination space (the direction of the gradient) [44] and the strength of the correlation between the ordination and environmental variable (the strength of the gradient). The vector fitting and *p* values were obtained with the function envfit from the vegan library [43], and significant explanatory variables (*p* < 0.05) were selected. For the different classes of macroinvertebrates, a random forest regression model was used to analyse the relative importance of environmental factors to their density changes.

The random forest regression model analysis was completed using the R package ‘randomForest’. The R package ‘samba’ was used to test whether the slope and intercept difference between the regression equations were significant [45].

## 3. Results

### 3.1. Changes in Nutrient Concentrations in the Water Column and Sediments in North Taihu

The TN in North Taihu showed a significant declining trend from 4.79 mg·L^−1^ in 2007 to 2.48 mg·L^−1^ in 2019 (MK test, *p* < 0.05). TP first declined from 1.53 mg·L^−1^ in 2007 to 1.25 mg·L^−1^ in 2012 and then began to rebound (Figure 2). From 2007 to 2012, Chla in North Taihu also declined but then began to rebound, and the Chla concentration in 2017 reached 67 μg·L^−1^ (Figure 2). The TLI eutrophication index showed that the eutrophication in northern Taihu first decreased and then increased, and the TLI multiyear average was 81.9 (Figure 2). The TN and TP in the sediment first increased and then decreased, and the turning point was in 2013. The multiyear average values of TNs and TPs were 1858 mg·kg^−1^ and 528 mg·kg^−1^, respectively, and the LOI showed an increasing but not significant trend (MK test, *p* > 0.05). The silt% maintained fluctuations of approximately 78% after 2009 but increased significantly from 65.6% in 2007 to 77% in 2009. Affected by massive stocking, the fish yield of Taihu increased year by year (MK test, *p* < 0.01).

### 3.2. Succession in Macroinvertebrate Community Structure in North Taihu

The number of species and dominant species of benthic animals in Taihu Lake did not change significantly from 2007 to 2019 (Appendix A). Turning point analysis shows the oligochaete density decreased from 2007 to 2013 and has increased since 2013 (Figure 3a) (turning points 2013). In 2007, among the sampling points in North Taihu, the highest density of oligochaetes was 78,840 ind·m^−2^. Turning point analysis shows the bivalves in North Taihu gradually increased before 2013 (turning points 2013), and the density remained stable from 2013 to 2019 (Figure 3b). The annual average density of gastropods in the northern lake area showed an increasing trend during the sampling period, from 9.1 ind·m^−2^ in 2007 to 25.9 ind·m^−2^ in 2019 (Figure 3c). The density of Polychaeta in the northern lake region first increased and then decreased, and the multiyear mean density was 81 ind·m^−2^ (Figure 3d). Crustacean density has declined each year in recent years (Figure 3e). From 2007 to 2017, the chironomid larvae in the northern lake area first decreased and then increased, and the density declined in the last two years (Figure 3e).

NMDS analysis (stress = 0.18) showed that the community structure in the northern lake area had a horseshoe shape (Figure 4). The significant factors driving the change in community structure were the TPs, TNs, fish yield, wind speed (WS), and LOI.

### 3.3. Seasonal Variation in Diversity Index of Macroinvertebrate

The high-frequency monitoring results of bottom-layer dissolved oxygen (DO%) also indicate that the occurrence of low dissolved oxygen concentrations increased in August after 2013. However, there were no significant changes observed in May. Further analysis showed that after 2013, in the autumn and winter seasons (November and February) without blooms, the diversity of macroinvertebrates in North Taihu did not show a declining trend, while in the season of cyanobacterial blooms, the diversity index showed a declining trend (Figure 5).

### 3.4. The Relationship between Macroinvertebrates and Trophic Level

According to the characteristics of the TP and Chla of North Taihu starting to rebound after 2013, the 2007–2019 period is divided into two different stages. During the period 2007–2013, there was a good correlation between the number of oligochaetes and the TLI index (r^2^ = 0.50, *p* < 0.001), but during the period 2014–2019, although the two were still significantly correlated (r^2^ = 0.14, *p* < 0.01), the correlation slope was significantly lower (*p* < 0.001) than that in the previous period, and the intercept of the regression equation was also significantly different (*p* = 0.007) (Figure 6 and Appendix A). Similarly, during the period 2007–2013, the Shannon–Wiener index and the TLI index showed a significant correlation (r^2^ = 0.30, *p* < 0.001), but during the period 2014–2019, the linear regression results showed that the relationship between the two was not significant. The Margalef diversity index and Pielou’s index also showed the same results (Appendix A).

## 4. Discussion

### 4.1. Intensified Internal Loading in Taihu

From 2007 to the present, there was no significant increase in external nutrient loading in Taihu [46], but eutrophication rebounded after 2013. This should be attributed to the increasing importance of internal loading. Our results show that while eutrophication rebounded, the nutrient concentration of surface sediments began to decline. Sediment nutrient changes are mainly affected by internal loading and external inputs; in addition, large-scale dredging activities may cause changes in sediment nutrients. In northern Taihu, the TN and TP in the sediments began to decrease after 2013. Most of the dredging activities in Taihu occurred before 2013, and the dredging area was relatively small (Figure 7); therefore, dredging was unlikely to be the main cause of nutrient changes in the sediments. Xu et al. [46] also showed that, in Taihu, a considerable part of the nutrient input from rivers is still buried in the sediments, especially phosphorus. The sedimentation rates of Taihu are approximately 0.6–3.6 mm per year (average 2 mm per year) [30], and the lacustrine sediment depth of Taihu is approximately 0.5–2 m [30]. The surface sediment sampling depth is approximately 5–10 cm, so the deposition rate has a limited impact on the nutrient concentration of the surface sediments within a few years. A possible explanation is that the nutrients in the sediments were released into the water.

### 4.2. Changes in the Community Structure of Macroinvertebrates and Their Influencing Factors

Previous studies have shown that lake macroinvertebrates are sensitive to environmental changes and are strongly influenced by oxygen, predation, substratum type, and habitat complexity [47,48,49,50]. Our results show that sediment indicators (TNs, TPs, LOI, and silt%), fish predation pressure, and WS are significant factors affecting the changes in the community structure of macroinvertebrates in North Taihu. TPs, TNs, and LOI indicate the nutritional and organic matter status of sediments. For macroinvertebrates in eutrophic lakes, hypoxia may be more important than organic enrichment. Although Taihu has strong wind disturbances, high organic matter means that even a short calm at the bottom of the lake will produce anaerobic bacteria, especially in the cyanobacteria bloom season, and cyanobacteria decompose and quickly consume oxygen [23]. With increasing fish stocking year by year, their predation effect on macroinvertebrates will increase. Despite the surprisingly high fish yield, the number of Polychaeta has increased in recent years, and random forest models also show that the predation pressure of fish is not the most important effect on macroinvertebrates (Appendix A). This may be because piscivorous fish and benthivorous fish account for a relatively small proportion of the fish populations in Taihu (approximately 11.9%) [51].

### 4.3. Macroinvertebrate Response to Internal Loading Changes

In Taihu, under the situation of an increase in internal loading, although the eutrophication of lakes has increased, there is no significant increase in nutrient input. During the cyanobacteria outbreak season, the death and decomposition of algae can easily lead to hypoxia on the sediment surface, and many benthic macroinvertebrates will suffocate [52,53]; however, in the cold season, macroinvertebrates will not be disturbed by more pollutants. For multiyear changes, these two effects are superimposed on macroinvertebrates in different seasons in one year. This difference led to the difference in the sensitivity of the oligochaete and Shannon–Wiener diversity index vs. the TLI index before and after 2013 (significant difference in the slope of the regression equation).

### 4.4. Impact on the Use of Lake Macroinvertebrates for Eutrophication Assessment

Regarding the use of macroinvertebrates for eutrophication assessment, our results showed that from community to population, macroinvertebrates responded differently to eutrophication changes caused by internal loading or external input. Especially for the oligochaete number and the diversity index, most of the assessment methods consider their response to eutrophication [37,54,55]. Under the current scenario of increased internal loading, the previous evaluation methods may lose their sensitivity.

## 5. Conclusions

With the gradual control of external nutrient loading, the contribution of internal loading to eutrophication in recent years has gradually increased. Our results show that the community structure of macroinvertebrates responds markedly differently to this. Intensified internal loading leads to weaker (or not significant) correlations between macroinvertebrate diversity and eutrophication-tolerant species numbers with the TLI index. The main reason for this change is that during internal loading intensification, the DO at the water-sediment interface improves in seasons without blooms, whereas hypoxia occurs frequently during blooms (Figure 8). By comparing the two lake survey results in 2008 and 2018 in the EPL, the results show that compared with 2008, lakes N and P did not increase significantly, but the Chla concentration of most lakes increased. The relationship between macroinvertebrates and TLI also showed similar results to Lake Taihu.

## Figures and Tables

**Figure 1 biology-12-01247-f001:**
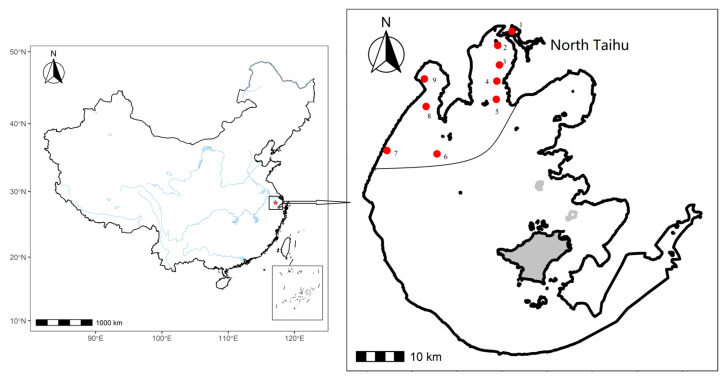
Study area and sampling sites in Lake Taihu.

**Figure 2 biology-12-01247-f002:**
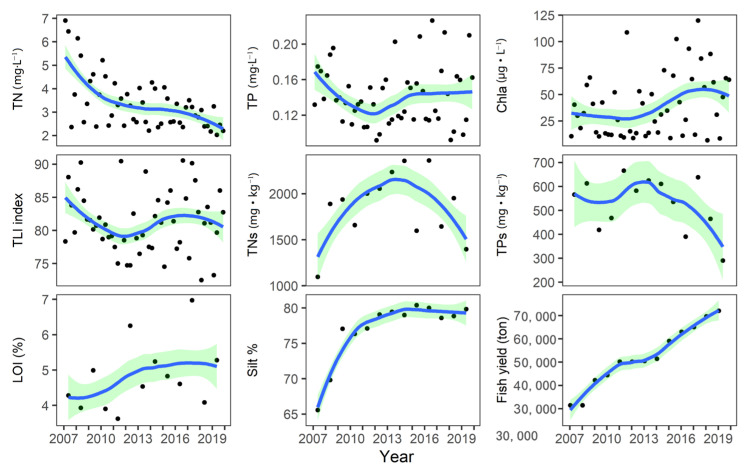
Changes in nutrient concentrations in the water column and sediment and chlorophyll concentrations in Lake Taihu from 2007 to 2019. (The data are converted to a power of 0.25, and the green range represents the 95% confidence interval. TN: total nitrogen, TP: total phosphorus Chla: chlorophyll, LOI: loss on ignition, TNs: sediment total nitrogen, TPs: sediment total phosphorus, TLI: trophic level index, % silt: percentage of silt).

**Figure 3 biology-12-01247-f003:**
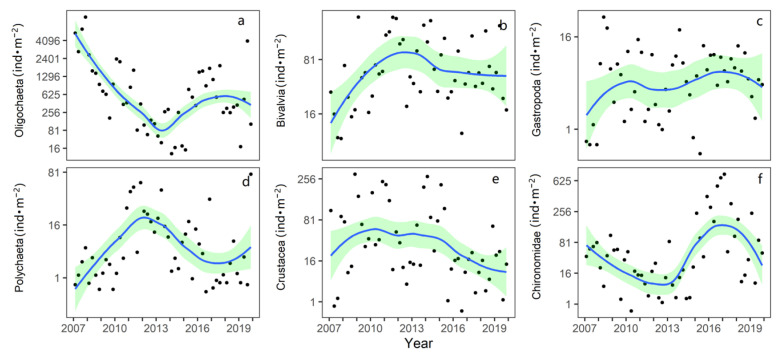
Changes in the abundance of macroinvertebrates for different classes in North Taihu from 2007 to 2019. (The data represents the average values from 9 sampling sites for each season and converted to a power of 0.25, and the green range represents the 95% confidence interval. (**a**–**f**) respectively represent the density change of Oligochaetes, Bivalvia, Gastropoda, Polychaeta, Crustacean, Chironomidae).

**Figure 4 biology-12-01247-f004:**
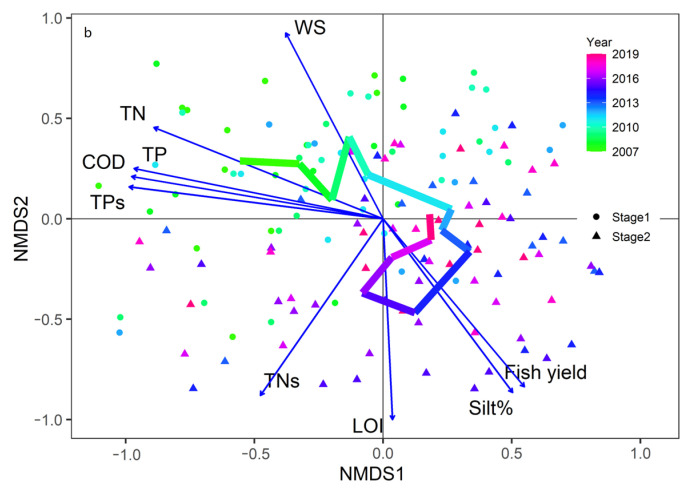
Changes in the community structure of macroinvertebrates in Taihu in two stages and significant environmental factors affecting their changes. The line represents the annual average change in the sample site score in NMDS analyses. (Stage 1: 2007–2013, Stage 2: 2014–2019. TN: total nitrogen, TP: total phosphorus, LOI: loss on ignition, TNs: sediment total nitrogen, TPs: sediment total phosphorus, % silt: percentage of silt, WS: wind speed).

**Figure 5 biology-12-01247-f005:**
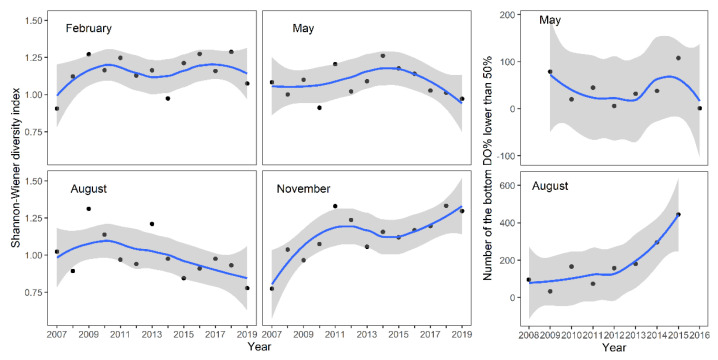
Changes in the benthic diversity index of Taihu in different seasons from 2007 to 2019 and the number of times that the DO% was lower than 50% in the hourly DO high-frequency monitoring results. (No bottom dissolved oxygen value was less than 50% in February and November. The grey range represents the 95% confidence interval).

**Figure 6 biology-12-01247-f006:**
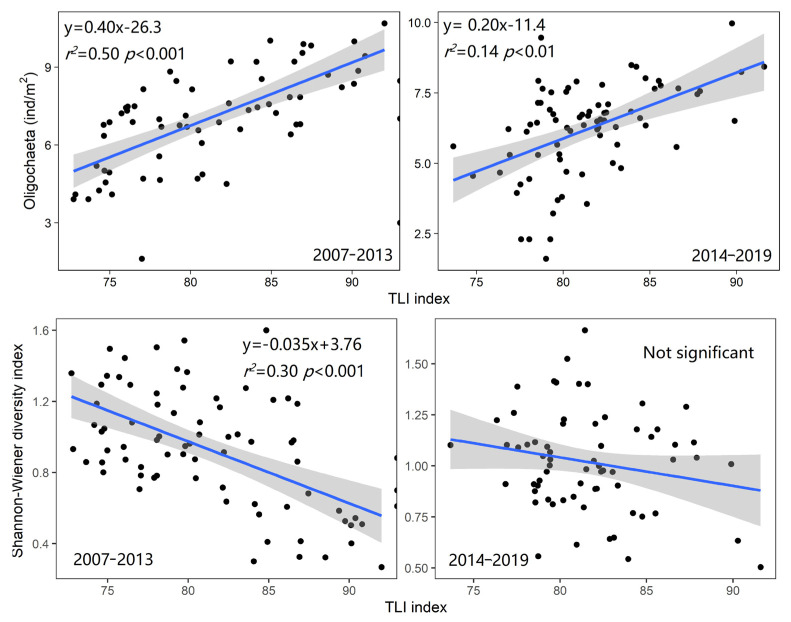
The relationship between the density of all oligochaetes, the Shannon–Wiener index, and the TLI index in northern Lake Taihu in different periods. (The Oligochaeta density is converted to the fourth power of the root sign, and the grey range represents the 95% confidence interval).

**Figure 7 biology-12-01247-f007:**
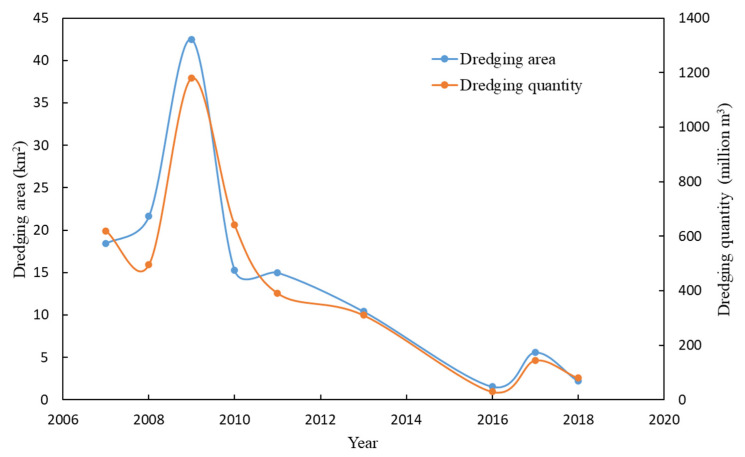
The dredging area and dredging quantity in Taihu from 2007 to 2018 (North Taihu Lake and East Taihu Lake are the main areas for ecological dredging).

**Figure 8 biology-12-01247-f008:**
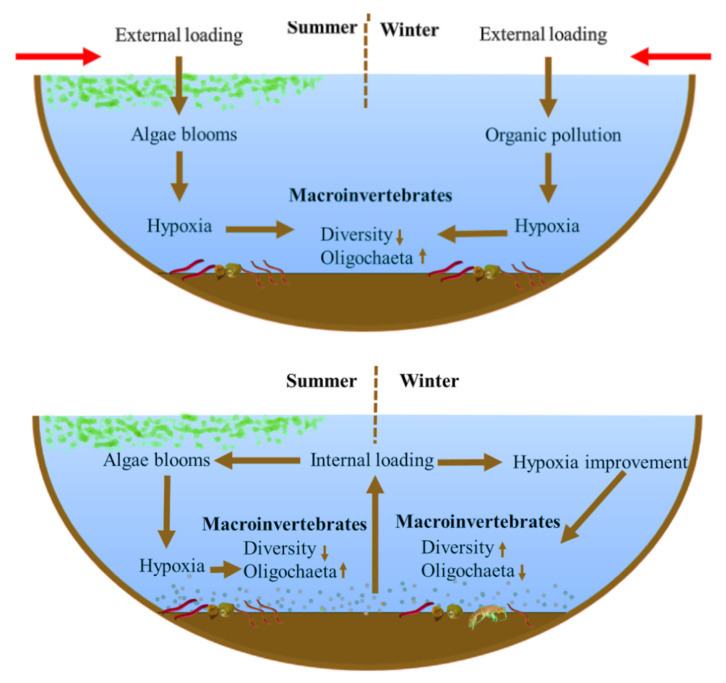
Differences in the response of macroinvertebrates to lake eutrophication dominated by external loading and internal loading.

## Data Availability

Not applicable.

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
