# Peer review of "Macroinvertebrate Response to Internal Nutrient Loading Increases in Shallow Eutrophic Lakes"

_biology, 2023, doi:10.3390/biology12091247_

Round 1
Reviewer 1 Report
Dear authors,
the manuscript entitled "Macroinvertebrate response to internal nutrient loading increases in shallow eutrophic lakes" aims to analyse how macroinvertebrates respond during eutrophication dominated by external input and internal loading. Overall, the manuscript is comprehensive and it clearly describes the methodology used to address the main goal.
Attached you can find some suggestions and comments.

Author Response
#Reviewer 1
Thank you for your comments. We have made the necessary revisions to the manuscript based on your suggestions, and we have directly addressed the specific errors within the manuscript.
Comments 1: It is difficult for me to follow these two sentences. You mentioned that there is a homogeneity in lake and the macroinvertebrates are similar among 9 sites. However, then you refer that among them, you selected 9 sampling sites. Please rephrase the sentences.
Response: We wanted to express that the northern region of Lake Taihu is characterized by severe eutrophication, with benthos predominantly composed of species tolerant to eutrophic. We realized that this statement is contradictory, so we have removed the first sentence.
We made the following changes:
“The sediment and water quality in North Taihu have strong spatial homogeneity, and the macroinvertebrate community structure and density between 9 different sampling sites are similar. Among them, 9 sampling sites were selected in North Taihu. The sampling times were February, May, August, and November of 2007–2019 (representing winter, spring, summer, and autumn, respectively).”

Reviewer 2 Report
The manuscript "Macroinvertebrate response to internal nutrient loading increases in shallow eutrophic lakes" by Kai Peng , Rui Dong , Boqiang Qin , Yongjiu Cai * , Jianming Deng , Zhijun Gong is of great ecological and practical value.
I read it with a great interest! The authors made huge work and I'm amazed by their R analysis skills.
I definitely recommend this manuscript for publication.
However, I have a couple of comments:
1) In Materials and methods you mentioned "28 shallow (mean depths between 1.5 and 9.0 m) lakes were selected as the research objects, most of which are eutrophic lakes " , I think Figure S1 and Table S1 should be placed in the main text.
2) Please provide list of species you identified in the Lake
3) In Introduction section add paragraph on comparison the divestity in other lakes.
4) In discussion section give a paragraph discussing the diversity/ oligochaeta phenomenon in other Lakes, not onle Taihu
5) In Figure 8 please change the colour of the macroinvertebrates - it is difficult to identify them on brown background. And provide a legend. I can see only some Oligochaeta, Chironomida and Molluscs? Arrows also would be better in different color. Algal bloom and the level of hypoxia impact on water transparency. I think the difference in water transparency should also be demonstrated ot the scheme.
That's all my comments.
Author Response
#Reviewer 2
Comments 2 In Materials and methods you mentioned "28 shallow (mean depths between 1.5 and 9.0 m) lakes were selected as the research objects, most of which are eutrophic lakes" , I think Figure S1 and Table S1 should be placed in the main text.
Response: Done!
Comments 3 Please provide list of species you identified in the Lake.
Response: Based on the feedback from other reviewers, we have included a description of the species classification information in the manuscript. Additionally, we have provided a list of the top 20 dominant species used in the manuscript.
Comments 4 In Introduction section add paragraph on comparison the divestity in other lakes.
Response: Based on the feedback from other reviewers, we have removed other lakes results in the manuscript.
Comments 5 In discussion section give a paragraph discussing the diversity/ oligochaeta phenomenon in other Lakes, not onle Taihu
Response: Based on the feedback from other reviewers, we have removed other lakes results in the manuscript.
Comments 6 In Figure 8 please change the colour of the macroinvertebrates - it is difficult to identify them on brown background. And provide a legend. I can see only some Oligochaeta, Chironomida and Molluscs? Arrows also would be better in different color. Algal bloom and the level of hypoxia impact on water transparency. I think the difference in water transparency should also be demonstrated ot the scheme.
Response: We have recolored Figure 8, which may provide better clarity.
Reviewer 3 Report
The manuscript brings an interesting and important subject, and it is overall well written and well organized.
I have a question about the justification of this study. On line 63 authors say “Despite this, few studies have investigated changes in macroinvertebrates when internal loading is increased.”
I wonder why macroinvertebrates should respond to internal loading because as I understand it, internal loading is about nutrients going from the sediment to the water column. However, macroinvertebrates are always in the sediment, adapted to the nutritional conditions found there. Therefore, the internal loading in the water column should not affect them as much.
Another main issue; an important event for the whole study was the eutrophication rebound that occurred in 2013 supposedly because internal loading. However, it is unclear what events and processes that led to this (i.e., which factors led to internal loading). Given the importance of this event for the work, it should be better explored and explained.
Other comments:
Lines 60-62: “Benthic macroinvertebrates live on the surfaces of sediments, and changes in the physical and chemical properties of sediments impact them [17].”
This phrase should go in the beginning of the paragraph.
Lines 121-23: “The sediment and water quality in North Taihu have strong spatial homogeneity, and the macroinvertebrate community structure and density between 9 different sampling sites are similar”
Is this a result of this study or a previous knowledge? If so, give the reference of the study.
Lines 193-195: “We used canonical correspondence analysis (CCA) and nonmetric multidimensional scaling (NMDS) analysis to show the characteristics of macroinvertebrate community changes.”
Authors should use constrained and unconstrained ordinations that complement themselves, in this way facilitating mutual comparisons. The unconstrained version of CCA is CA (correspondence analysis), more specifically DCA (detrended correspondence analysis).
Lines 202-213: Then the authors used envifit to see the correlation of the NMDS axes with the environmental variables. Is this not redundant with CCA, which has a similar purpose of relating compositional gradients to environmental variables?
Line 213: What are the “classes” of macroinvertebrates?
Line 220: In Results authors should present measures of variation of the sediment metrics among the sites, in this way confirming (or not) the previous claim of homogeneity in sediment conditions.
Line 248: Which criteria was used to species to be considered dominant?
Lines 264-265: “The environmental factors that significantly affected the community structure screened by the two analysis methods were also roughly similar.”
This is expected. As I mentioned before, these two statistical approaches are somehow redundant.
Lines 365-366: “Parametric analysis (e.g., CCA) is sensitive to the absolute number of species, while nonparametric analysis is not.”
Number of species or number of individuals?
Anyway, who said that??
The properties of the NMDS are largely dependent on the properties of the dissimilarity measure used. In this case, the authors used the Bray–Curtis index, which is greatly influenced by the number of species/individuals!
Line 402: Authors should not end the Conclusion presenting results (“By comparing the two lake survey results in 2008 and 2018 in the EPL, the results show that compared with 2008, lake N and P did not increase significantly, but the Chla concentration of most lakes increased. The relationship between macroinvertebrates and TLI also showed similar results to Lake Taihu.”). The conclusion of the work should give the main message of the work, its innovation and how its results generally apply to lake biomonitoring using macroinvertebrates.
Nothing important.
Author Response
#Reviewer 3
Comments 7 I have a question about the justification of this study. On line 63 authors say “Despite this, few studies have investigated changes in macroinvertebrates when internal loading is increased.”
I wonder why macroinvertebrates should respond to internal loading because as I understand it, internal loading is about nutrients going from the sediment to the water column. However, macroinvertebrates are always in the sediment, adapted to the nutritional conditions found there. Therefore, the internal loading in the water column should not affect them as much.
Response: Yes, we agree with you. The intensification of internal loading is unlikely to have a direct impact on macroinvertebrates. What we want to emphasize is the response of macroinvertebrates to eutrophication caused by the intensification of internal loading. The main argument of this manuscript is also that the eutrophication resulting from intensified internal loading leads to seasonal variations in dissolved oxygen in sediments, which in turn affects macroinvertebrates differently.
We made the following changes:
“Despite this, few studies have investigated changes in macroinvertebrates when internal loading is increased. Increased internal loading leading to eutrophication will result in hypoxia at the water-sediment interface during algal bloom seasons. However, during non-algal bloom seasons, the concentration of dissolved oxygen at the water-sediment interface will improve due to the control of external pollutants.”
Comments 8 Another main issue; an important event for the whole study was the eutrophication rebound that occurred in 2013 supposedly because internal loading. However, it is unclear what events and processes that led to this (i.e., which factors led to internal loading). Given the importance of this event for the work, it should be better explored and explained.
Response: Many studies have indicated that the eutrophication rebound in Lake Taihu in 2013 was driven by climate change. For instance, significant changes in temperature, wind speed, and rainfall occurred around the period of 2013. We mentioned the following in lines 84-85, Previous studies have shown that this may be caused by climate change[1, 2].
We made the following changes: Previous studies have shown that this may be caused by climate change[1, 2]. For example, increases in temperature and decreases in wind speed can both contribute to the intensification of internal loading in Lake Taihu.
Comments 9 Lines 60-62: “Benthic macroinvertebrates live on the surfaces of sediments, and changes in the physical and chemical properties of sediments impact them [17].”
This phrase should go in the beginning of the paragraph.
Response: Done
Comments 10 Lines 121-23: “The sediment and water quality in North Taihu have strong spatial homogeneity, and the macroinvertebrate community structure and density between 9 different sampling sites are similar”
Is this a result of this study or a previous knowledge? If so, give the reference of the study.
Response: See comment 1
Comments 11 -16
Lines 193-195: “We used canonical correspondence analysis (CCA) and nonmetric multidimensional scaling (NMDS) analysis to show the characteristics of macroinvertebrate community changes.”
Authors should use constrained and unconstrained ordinations that complement themselves, in this way facilitating mutual comparisons. The unconstrained version of CCA is CA (correspondence analysis), more specifically DCA (detrended correspondence analysis).Lines 202-213: Then the authors used envifit to see the correlation of the NMDS axes with the environmental variables. Is this not redundant with CCA, which has a similar purpose of relating compositional gradients to environmental variables?
Line 220: In Results authors should present measures of variation of the sediment metrics among the sites, in this way confirming (or not) the previous claim of homogeneity in sediment conditions.
Lines 264-265: “The environmental factors that significantly affected the community structure screened by the two analysis methods were also roughly similar.”
This is expected. As I mentioned before, these two statistical approaches are somehow redundant.
Lines 365-366: “Parametric analysis (e.g., CCA) is sensitive to the absolute number of species, while nonparametric analysis is not.”
Number of species or number of individuals?
Anyway, who said that??
The properties of the NMDS are largely dependent on the properties of the dissimilarity measure used. In this case, the authors used the Bray–Curtis index, which is greatly influenced by the number of species/individuals!
Response: Based on your advice, we have chosen to retain the results of the NMDS analysis and exclude the CCA analysis results. We have also removed the comparison between the two analysis methods from the original text.
Comments 17 Line 213: What are the “classes” of macroinvertebrates?
Response: For the different classes(Oligochaete, Chironomus, Bivalvia, Gastropoda, Polychaeta, and Crustacea) of macroinvertebrates
Comments 18 Line 248: Which criteria was used to species to be considered dominant?
Response: We calculate the dominance of all species before and after 2013 for comparison. According to Tables S1, for the periods 2007-2013 and 2014-2019, there were seven dominant species among the top ten that did not undergo any changes in dominance rank.
Comments 18 Line 402: Authors should not end the Conclusion presenting results (“By comparing the two lake survey results in 2008 and 2018 in the EPL, the results show that compared with 2008, lake N and P did not increase significantly, but the Chla concentration of most lakes increased. The relationship between macroinvertebrates and TLI also showed similar results to Lake Taihu.”). The conclusion of the work should give the main message of the work, its innovation and how its results generally apply to lake biomonitoring using macroinvertebrates.
Response: Following the suggestion of Reviewer 4, the research results regarding the EPL region have been removed from the manuscript.
Reviewer 4 Report
The argument of macroinvertebrate response to internal nutrient loads in shallow eutrophic lakes is very interesting, especially in the era of globally increased water quality demand. Despite the importance of the argument, the MS needs to be improved. There are two critical points to be considered: (i) the results of 28 shallow lakes in the EPL are to be excluded/ if presented they should be comparable with the results of Lake Taihu in order to confirm your hypothesis, and (ii) the presentation of the statistical analysis (to be refine). Some suggestions and comments are provided to the authors for improvement of the MS:
1. Introduction
- Line 70-73: replace “By analysing the changes in macroinvertebrates to eutrophication dominated by internal loading, it is not only helpful to understand the response of lake ecosystems to eutrophication dominated by internal release but can also evaluate the effectiveness of eutrophication assessment based on macroinvertebrates” with “By analysing the changes in macroinvertebrates to eutrophication dominated by internal loading, it is not only helpful to understand the response of lake ecosystems to eutrophication but can also evaluate the effectiveness of eutrophication assessment based on macroinvertebrates”;
- Line 80: replace “in 2006” with “since 2006”;
- Line 85: cite reference for the phrase Previous studies have shown that this may be caused by climate change”;
- Line 105-106: Formulate better the second hypothesis, try to be more clear;
- Line 107-112: I suggest to exclude this part. The results of the 28 lakes could be perfect for another paper. The inclusion of this part just make more confused your findings. More, in this case are missing the parameters discussed for Lake Taihu such as total nitrogen and phosphorus in the sediments. Further, the EPL lakes are characterized by aquatic plants, while Lake Taihu seems not to present this characteristics. Submerged plants are important factor that could modified the results and contribute to dissimilarity of EPL lakes and Lake Taihu. The EPL lakes are also deeper than Lake Taihu.
2. Materials and methods:
- Fig.1S needs to be included in the main text of the paper and omitted in the Supplementary materials. It should be better presented, with legend, with number of the sampling station indicated on the map;
- Indicate the max depth of Lake Taihu;
- For conformity use through the whole text the same wording- “water-sediment interface” or “sediment-water interface”;
- Line 122-124: Clarify or correct the text regarding the number of sampling sites. In total are 9 sites and how many were selected?
- Line 136: Explain what kind of lake changes we mean on the phrase “The samples were collected in two different periods (a few sites could not be reached due to lake changes, so samples were taken nearby”;
- Explain how many replicates per sampling site we have sampled. 2 or 3?
- If you decide to include Fish yield data to justify the fish predation pressure, you should give more information about the fish species included in these data. Are they all species presented in the lake? You should exclude the planktivorous species but even in that way it is not the only predation pressure on macroinvertebrates. Finally, all conclusion do not confirm the significance od this factor so probably to exclude;
- Line 170: You should refer abbreviation of Mann-Kendall as MK just because you used KK through the text (e.g. line 222);
- Line 198: Explain why you have selected 20 dominant species? With how much % in the total abundance they were presented. In Table S2 you have include 10 dominant species of different groups. Explain the divergence between 10 in the supplementary and 20 dominant species in the section “Materials and methods”. Modify Table S2 and explain Species number is the total number of species? From taxonomical point of view, it will be better to present the taxonomy of the recorded species at least for dominant species. How many between the dominants are bivalves, or oligochaetes, or crustaceans. Why you have included only macroinvertebrates only for May if you have sampled also in other three months?
- For the turning point in the time-series you must use in R the function turnpoints in pastec. In this way you could be able to confirm your statement for turning point in 2013 and the determination of two periods 2007-2013 and 2014-2019.
- Residuals and Goodness-of-fit for the regression analysis are missing. You should perform residual analysis and plot the residuals to confirm the significance of your regression results;
3. Results:
- Replace all mg/L with mg.L-1 thorough the text
- Line 227: “ turning point was in 2013”- you should preform an appropriate statistical analysis;
- Line 228: correct “mg/kg”, use the correct unit;
- Missing the citation of Figure 1;
- Figure 1: correct the units, put the legend in the caption and give full description of the abbreviations LI, TP, TPs, etc.;
- Line 238-243: this part is somehow unclear because on Figure 2 are not evident your results “Compared with 2008, during the dry season, 16 lakes had decreased TN concentrations, 12 lakes had decreased TP concentrations, and 10 lakes had decreased Chla concentrations (Figure. 2a, b c). During the wet season, 13 lakes had decreased TP concentrations, 12 lakes had decreased TN concentrations, and the Chla concentration increased in almost all lakes (decreased in only one lake) (Figure 2d, e, f).” Make clear this statement, especially the number of lakes with decreased TN and Cla (is there any threshold?).
- Modify Fig.2 -bigger font, a-f are not readable, title of axis x (number of the lake according Table S1). You should put the ID of 28 lakes on the map (Figure 1). Why it is indicated 2009 and 2018 only if your studied period is 2007-2008? The values are in the water column and you have not presented results in the sediments?
- Line 261: replace “The community structure of the CCA” with “The community structure according to the CCA”;
- Line 262: again the turning point in 2013 could be identify by the appropriate analysis;
- Line 263: your results here that the Fish yield is a significant factor, in the Discussion is just stated the opposite. Please explain;
- Figure 3: Change the units, is this only the abundance in May, or it is an average of the four months samped? Make clear and explain;
- Figure 4: Make bigger the font. I suggest to call two stages periods. The colour legend is not clear. Why only five years are presented? Put legend of the factors in the caption;
- Figure 5 is not cited;
- Figure 5. Oligochaeta density is not clear of how many species is composed? Only the dominants? All oligochaetes? Only in May? The great dispersion and so low R2 could suggest low correlation between the variables in the regression analysis. You should try to see the residual plots and to check whether the observed error is consistent with the stochastic error (differences between the expected and observed values must be random and unpredictable) , otherwise your models seemed to not fit well (R2= 0.14). Just in case you have pooled monthly data, there could be seen the seasonal effect;
- Figure 6 does not confirm your statement of weakly correlation of TLI and diversity indices. Correct the units. Explain why in this case you converted oligochaetes density to fourth power of the root sign and in the previous case (Figure 5) not;
4. Discussion: The section is a mixed of results and discussion
- The citation and results presented in Fig S4 should be moved to the section Results;
- Line 332: check the reference number 52 dealing with the marine environments and not with eutrophicated lakes;
- Line 337-339: see the comment regrading the importance of the fish yield. Here you stated that it is not important, but in the results is just the opposite;
- Figure 7: move the figure to the results;
- Line 365-366: make clear your statement “Parametric analysis (e.g., CCA) is sensitive to the absolute number of species, while nonparametric analysis is not. Therefore, in the CCA, the community structure of benthic macroinvertebrates at different stages had obvious differences, while the NMDS analysis had relatively small differences in community structure.” Explain better;
- Line 382: explain what does mean “Due to the removal of nitrogen and phosphorus by large algal colonies and suspended solids measured by EMC data [58], our measurement method did not do this;
Conclusions: Difficult to find the reason why you include the study of the 28 lakes in this paper, also because your statement “By comparing the two lake survey results in 2008 and 2018 in the EPL, the results show that compared with 2008, lake N and P did not increase significantly, but the Chla concentration of most lakes increased. The relationship between macroinvertebrates and TLI also showed similar results to Lake Taihu” did not contribute to the aim of the study.
References:
- Correct ref number 2, some letters are capital letter
Author Response
#Reviewer 4
Comments 18 The argument of macroinvertebrate response to internal nutrient loads in shallow eutrophic lakes is very interesting, especially in the era of globally increased water quality demand. Despite the importance of the argument, the MS needs to be improved. There are two critical points to be considered: (i) the results of 28 shallow lakes in the EPL are to be excluded/ if presented they should be comparable with the results of Lake Taihu in order to confirm your hypothesis, and (ii) the presentation of the statistical analysis (to be refine). Some suggestions and comments are provided to the authors for improvement of the MS:
Response: We have followed your advice and incorporated the suggestions from other reviewers to remove the relevant content about the EPL region from the manuscript. We will focus on making significant revisions to the statistical analysis in both the earlier and later sections of the manuscript, as suggested in the comments.
Comments 19Line 70-73: replace “By analysing the changes in macroinvertebrates to eutrophication dominated by internal loading, it is not only helpful to understand the response of lake ecosystems to eutrophication dominated by internal release but can also evaluate the effectiveness of eutrophication assessment based on macroinvertebrates” with “By analysing the changes in macroinvertebrates to eutrophication dominated by internal loading, it is not only helpful to understand the response of lake ecosystems to eutrophication but can also evaluate the effectiveness of eutrophication assessment based on macroinvertebrates”;
Response: Done
Comments 20 Line 80: replace “in 2006” with “since 2006”;
Response: Done
Comments 21 Line 85: cite reference for the phrase Previous studies have shown that this may be caused by climate change”;
Response: Done! See comment 8.
Comments 22 Line 105-106: Formulate better the second hypothesis, try to be more clear;
Response: We made the following changes:
“The results of using macroinvertebrates for eutrophication assessment will lose sensi-tivity. If hypothesis 1 is valid, then the sensitivity of using macroinvertebrates for eu-trophication assessment, based on their correlation with eutrophication, would likely be compromised.”
Comments 23 Line 107-112: I suggest to exclude this part. The results of the 28 lakes could be perfect for another paper. The inclusion of this part just make more confused your findings. More, in this case are missing the parameters discussed for Lake Taihu such as total nitrogen and phosphorus in the sediments. Further, the EPL lakes are characterized by aquatic plants, while Lake Taihu seems not to present this characteristics. Submerged plants are important factor that could modified the results and contribute to dissimilarity of EPL lakes and Lake Taihu. The EPL lakes are also deeper than Lake Taihu.
Response: Done! We exclude this part
Comments 23 Fig.1S needs to be included in the main text of the paper and omitted in the Supplementary materials. It should be better presented, with legend, with number of the sampling station indicated on the map;
Response: Done!
Comments 23 Indicate the max depth of Lake Taihu;
Response: Done!
Comments 24 For conformity use through the whole text the same wording- “water-sediment interface” or “sediment-water interface”
Response: Done!
Comments 25 Line 122-124: Clarify or correct the text regarding the number of sampling sites. In total are 9 sites and how many were selected?
Response: We have made modifications to the figure, showing only 9 monitoring sites. See figure 1.
Comments 25 Line 136: Explain what kind of lake changes we mean on the phrase “The samples were collected in two different periods (a few sites could not be reached due to lake changes, so samples were taken nearby”;
Response: We have removed the part related to the EPL region from the manuscript.
Comments 26 Explain how many replicates per sampling site we have sampled. 2 or 3?
Response: 3, we have revised.
Comments 26 If you decide to include Fish yield data to justify the fish predation pressure, you should give more information about the fish species included in these data. Are they all species presented in the lake? You should exclude the planktivorous species but even in that way it is not the only predation pressure on macroinvertebrates. Finally, all conclusion do not confirm the significance od this factor so probably to exclude;
Response: Yes, we agree with your point. Unfortunately, we only have fishery production data published by government agencies, and we do not have information on the species composition of fish communities. However, for macroinvertebrates, fish predation is an important influencing factor. Therefore, we can only use fishery production data as an approximation for fishing pressure on macroinvertebrates. The fishery production data is obtained through the statistical records of commercial fishing activities by government agencies, and thus, we believe it can be used as a proxy for predation pressure on macroinvertebrates.
Comments 27 You should refer abbreviation of Mann-Kendall as MK just because you used KK through the text (e.g. line 222);
Response: We made the following changes:
“Mann–Kendall trend analyses(MK test) were used to test their interannual trends. Mann–Kendall trend analyses MK test were carried out using the function “Kendall” in the R package Kendall [3].”
Comments 28 Line 198: Explain why you have selected 20 dominant species? With how much % in the total abundance they were presented. In Table S2 you have include 10 dominant species of different groups. Explain the divergence between 10 in the supplementary and 20 dominant species in the section “Materials and methods”. Modify Table S2 and explain Species number is the total number of species? From taxonomical point of view, it will be better to present the taxonomy of the recorded species at least for dominant species. How many between the dominants are bivalves, or oligochaetes, or crustaceans. Why you have included only macroinvertebrates only for May if you have sampled also in other three months?
Response: The top 20 dominant species were determined based on our expertise, with the 20th species accounting for 0.27% of the total number of species. We have made the recommended modifications to Table S1 and added classification information in the original manuscript, as per your suggestion. Additionally, we only sampled macroinvertebrates in the months of February, May, August, and November, while sediment sampling was conducted only in the month of May each year.
We made the following changes:
“In total, 52 taxa were recorded from the 9 sites during the eight years, including five polychaetes, seven oligochaetes, 11 aquatic insects (chironomids), 12 bivalves, 7 gastropods and 8 malacostraca.”
Comments 29 For the turning point in the time-series you must use in R the function turnpoints in pastec. In this way you could be able to confirm your statement for turning point in 2013 and the determination of two periods 2007-2013 and 2014-2019.
Response: Done!
We made the following changes:
“Turning point analysis were used to identify the year in which a significant change in macroinvertebrate density. Turning point analysis were carried out using the function “turnpoints” in the R package pastecs[37].”
Comments 30 Residuals and Goodness-of-fit for the regression analysis are missing. You should perform residual analysis and plot the residuals to confirm the significance of your regression results;
Response: Done!
We made the following changes:
“Linear regression is used to analyze the direct relationship between TLI index, macroinvertebrates diversity index, and oligochaete density. The Residuals vs Fitted plot is used to assess the reliability of the regression model.”
Comments 31 Replace all mg/L with mg.L-1 thorough the text
Response: Done!
Comments 32 Line 227: “turning point was in 2013”- you should preform an appropriate statistical analysis;
Response: Done!
Comments 33 Missing the citation of Figure 1;
Response: Done!
Comments 34 Figure 1: correct the units, put the legend in the caption and give full description of the abbreviations LI, TP, TPs, etc.;
Response: Done!
Comments 35 Line 238-243: this part is somehow unclear because on Figure 2 are not evident your results “Compared with 2008, during the dry season, 16 lakes had decreased TN concentrations, 12 lakes had decreased TP concentrations, and 10 lakes had decreased Chla concentrations (Figure. 2a, b c). During the wet season, 13 lakes had decreased TP concentrations, 12 lakes had decreased TN concentrations, and the Chla concentration increased in almost all lakes (decreased in only one lake) (Figure 2d, e, f).” Make clear this statement, especially the number of lakes with decreased TN and Cla (is there any threshold?).
Response: We have removed the part related to the EPL region from the manuscript.
Comments 36 Modify Fig.2 -bigger font, a-f are not readable, title of axis x (number of the lake according Table S1). You should put the ID of 28 lakes on the map (Figure 1). Why it is indicated 2009 and 2018 only if your studied period is 2007-2008? The values are in the water column and you have not presented results in the sediments?
Response: We have removed the part related to the EPL region from the manuscript.
Comments 37: Line 261: replace “The community structure of the CCA” with “The community structure according to the CCA”;
Response: According to the suggestion of Reviewer 3, we have removed the CCA analysis from the content.
Comments 38: Line 262: again the turning point in 2013 could be identify by the appropriate analysis;
Response: Done!
Comments 39: Line 263: your results here that the Fish yield is a significant factor, in the Discussion is just stated the opposite. Please explain;
Response: Done!
Comments 40: Figure 3: Change the units, is this only the abundance in May, or it is an average of the four months samped? Make clear and explain;
Response: Figure 3 presents data for the four seasons (spring, summer, autumn, and winter) from the years 2007 to 2019. We have revised.
Comments 41: Figure 4: Make bigger the font. I suggest to call two stages periods. The colour legend is not clear. Why only five years are presented? Put legend of the factors in the caption;
Response: Done! There are too many years to display in the legend, as the colors representing the years are continuous gradients. Therefore, we only included a subset of the years in the legend.
Comments 42: Figure 5 is not cited;
Response: we have cited in the manuscript.
Comments 43: Figure 5. Oligochaeta density is not clear of how many species is composed? Only the dominants? All oligochaetes? Only in May? The great dispersion and so low R2 could suggest low correlation between the variables in the regression analysis. You should try to see the residual plots and to check whether the observed error is consistent with the stochastic error (differences between the expected and observed values must be random and unpredictable) , otherwise your models seemed to not fit well (R2= 0.14). Just in case you have pooled monthly data, there could be seen the seasonal effect;
Response: Oligochaeta density include all 7 oligochaete species. The residual plot is shown below. We used the annual average values from each monitoring site for regression analysis, in order to avoid the influence of seasonal effects. Based on our results, the lower R-squared value is attributed to the changes in sediment interface dissolved oxygen concentration following intensified endogenous release, which resulted in a shift in macroinvertebrates response to eutrophication. As a result, the correlation weakened significantly after 2013.
|
b) |
|
a) |
|
d) |
|
c) |
Figure S1 The Residuals vs Fitted plot between the density of all oligochaetes, the Shannon–Wiener index and the TLI index in northern Lake Taihu in different periods. (a: oligochaetes VS TLI index during 2007-2013; b: oligochaetes VS TLI index during 2014-2019; c: Shannon–Wiener index VS TLI index during 2007-2013; d: Shannon–Wiener index VS TLI index during 2013-2019)
Comments 44: Figure 6 does not confirm your statement of weakly correlation of TLI and diversity indices. Correct the units. Explain why in this case you converted oligochaetes density to fourth power of the root sign and in the previous case (Figure 5) not;
Response: Based on the previous suggestion, we have removed this section.
Comments 45: The citation and results presented in Fig S4 should be moved to the section Results;
Response: Done!
Comments 46: Line 332: check the reference number 52 dealing with the marine environments and not with eutrophicated lakes;
Response: We have change

Reviewer 5 Report
I am not convinced of the element of novelty introduced by the assessed work. This element is not sufficiently emphasised in the individual chapters of the study.
Some general comments:
Surely eutrophication is a problem facing lakes? It is a natural process that occurs in nature. The main threat to lakes is the unnaturally rapid increase in lake trophicity caused by human activities.
Benthic organisms can live both on and in the sediment surface.
The occurrence of algal blooms is mainly caused by large fluctuations in oxygen concentration.
Is it really true that the effects of hypoxic bottom waters on benthic organisms are unknown? This statement is highly controversial. One can easily point to many papers on this subject.
Methodology:
Please explain your choice of biological sampling tool. The Petersen sampler has some limitations in terms of substrate type. Why was water sampled from the strata if the study included shallow water lakes, i.e. polymictic lakes? The statistical method does not indicate how the descriptor variables were selected.
Results:
I do not know what test is hidden behind the abbreviation MK. The models do not have descriptions of quality and how much explanation was provided by each axis, what the stress was, etc. The presentation of the results in the form of an ordination is not very convincing. Perhaps it would be worthwhile to use the p-ie technique, where the years could be presented and the main macrozoobenthos species exposed.
Author Response
#Reviewer 5
Comments 53: Surely eutrophication is a problem facing lakes? It is a natural process that occurs in nature. The main threat to lakes is the unnaturally rapid increase in lake trophicity caused by human activities.
Response: We made the following changes:
“Eutrophication is a critical worldwide environmental issue facing lake ecosystems. The excessive input of nutrients causes severe eutrophication of lakes. Excessive accumulation of nutrients resulting from human activities has led to eutrophication, posing significant threats to lake ecosystems.”
Comments 54: Benthic organisms can live both on and in the sediment surface.
Response: We made the following changes:
“Benthic macroinvertebrates mainly live on the surfaces of sediments, and changes in the physical and chemical properties of sediments impact them.”
Comments 55: The occurrence of algal blooms is mainly caused by large fluctuations in oxygen concentration.
Eutrophication affects the community structure, leading to a reduction in diversity and an increase in the number of pollution-tolerant species [5-7], in part due to a large fluctuations reduction in oxygen at the water-sediment interface
Comments 56: Is it really true that the effects of hypoxic bottom waters on benthic organisms are unknown? This statement is highly controversial. One can easily point to many papers on this subject.
Response: There have been numerous studies on the impact of changes in sediment interface dissolved oxygen on benthic organisms. What we want to convey is that intensified internal loading leading to eutrophication primarily causes a decrease in bottom layer dissolved oxygen during algal bloom seasons. However, during non-algal bloom seasons, the control of exogenous pollution prevents the deterioration of bottom layer dissolved oxygen. Additionally, the impact of intensified endogenous release and eutrophication on benthic organisms is relatively minimal.
We made the following changes:
“Despite this, few studies have investigated changes in macroinvertebrates when in-ternal loading is increased. Increased internal loading leading to eutrophication will result in hypoxia at the water-sediment interface during algal bloom seasons. However, during non-algal bloom seasons, the concentration of dissolved oxygen at the water-sediment interface will improve due to the control of exogenous pollutants.”
Methodology:
Comments 57: Please explain your choice of biological sampling tool. The Petersen sampler has some limitations in terms of substrate type. Why was water sampled from the strata if the study included shallow water lakes, i.e. polymictic lakes? The statistical method does not indicate how the descriptor variables were selected.
As mentioned, the sediments in Lake Taihu mainly consist of silt, and the substrate type is relatively homogeneous. Therefore, we used the Peterson sampler for sampling. As for the lakes in the EPL region, we have removed the content according to the suggestions of the previous reviewers.
Results:
Comments 58: I do not know what test is hidden behind the abbreviation MK. The models do not have descriptions of quality and how much explanation was provided by each axis, what the stress was, etc. The presentation of the results in the form of an ordination is not very convincing. Perhaps it would be worthwhile to use the p-ie technique, where the years could be presented and the main macrozoobenthos species exposed.
MK means Mann–Kendall trend analyses, we have revised. In the NMDS analysis, we performed factor selection using the "envfit" function to identify significant environmental factors that influence the data. Regarding the statistical analysis, we have made significant revisions based on the suggestions of the previous reviewers. We have also included additional information such as stress values.

Round 2
Reviewer 4 Report
The authors have taken in consideration the suggestions and comments.
Author Response
Thank you
Reviewer 5 Report
Figure 1 needs to be refined. A wider area, such as China, should be added to define the location of the lake under study.
Author Response
Done!
